# Deep learning for classification of pediatric chest radiographs by WHO's standardized methodology

**Yiyun Chen**[1]*, **Craig S. Roberts**[1], **Wanmei Ou**[1], **Tanaz Petigara**[1], **Gregory V. Goldmacher**[1], **Nicholas Fancourt**[2], **Maria Deloria Knoll**[3]

**1** Merck & Co., Inc., Kenilworth, New Jersey, United States of America, **2** Menzies School of Health Research, Charles Darwin University, Darwin, Australia, **3** Department of International Health, International Vaccine Access Center, Johns Hopkins Bloomberg School of Public Health, Baltimore, Maryland, United States of America

* star.chen@merck.com

## Abstract

### Background

The World Health Organization (WHO)-defined radiological pneumonia is a preferred end-point in pneumococcal vaccine efficacy and effectiveness studies in children. Automating the WHO methodology may support more widespread application of this endpoint.

### Methods

We trained a deep learning model to classify pneumonia CXRs in children using the World Health Organization (WHO)'s standardized methodology. The model was pretrained on CheXpert, a dataset containing 224,316 adult CXRs, and fine-tuned on PERCH, a pediatric dataset containing 4,172 CXRs. The model was then tested on two pediatric CXR datasets released by WHO. We also compared the model's performance to that of radiologists and pediatricians.

### Results

The average area under the receiver operating characteristic curve (AUC) for primary end-point pneumonia (PEP) across 10-fold validation of PERCH images was 0.928; average AUC after testing on WHO images was 0.977. The model's classification performance was better on test images with high inter-observer agreement; however, the model still outperformed human assessments in AUC and precision-recall spaces on low agreement images.

### Conclusion

A deep learning model can classify pneumonia CXR images in children at a performance comparable to human readers. Our method lays a strong foundation for the potential inclusion of computer-aided readings of pediatric CXRs in vaccine trials and epidemiology studies.

**Data Availability Statement:** The PERCH data contain potentially sensitive information and are not yet publicly available. The data are currently owned by the Johns Hopkins University and the

PERCH Consortium. Request to access the data can be sent to Christine Prosperi (cprospe1@jhu.edu). Source code implementing all steps—data preprocessing, training, testing and visualization—is available from https://github.com/KookyGhost/PERCH_CXR.git. The CheXpert dataset is publicly available and can be downloaded from: https://stanfordmlgroup.github.io/competitions/chexpert/. The WHO dataset is publicly available and can be obtained through the WHO radiology working group: medicaldevices@who.int.

**Funding:** The PERCH study was supported by grant 48968 from the Bill & Melinda Gates Foundation (https://www.gatesfoundation.org/) to the International Vaccine Access Center, Department of International Health, Johns Hopkins Bloomberg School of Public Health (Baltimore, MD, USA). Maria Deloria Knoll received a small grant from Merck & Co. to cover expenses related to preparing the PERCH dataset shared for use in the study, and for consulting on the manuscript. Yiyun Chen, Tanaz Petigara, Wanmei Ou, Craig S. Robert, Gregory V. Goldmacher were employees of Merck & Co., Inc. during the conduct of the study. Nicholas Fancourt has no affiliations with or involvement in any organization or entity with any financial interest.

**Competing interests:** Yiyun Chen, Tanaz Petigara, Wanmei Ou, Craig S. Robert, Gregory V. Goldmacher were employees of Merck & Co., Inc. during the conduct of the study. Maria Deloria Knoll received grant from Merck & Co. to cover expenses related to preparing the PERCH dataset shared for use in the study, and for consulting on the manuscripts. The commercial affiliation does not alter our adherence to PLOS ONE policies on sharing data and materials.

## Introduction

Pneumonia is the leading infectious cause of death in children. *Streptococcus pneumoniae*, a gram positive bacterium, is a common cause of bacterial pneumonia in children. In 2015, almost 300,000 deaths due to pneumococcal pneumonia were estimated to have occurred in children less than 5 years, primarily in Africa and Asia [1].

Pneumococcal conjugate vaccines (PCV) are highly effective at preventing pneumococcal disease [2], but estimating their impact on pneumonia in children requires large studies and standardized case definitions. Currently available biological tests are insufficient to identify the etiology of pneumonia in children; as antigen tests lack specificity, blood cultures lack sensitivity, and lung aspirates are impractical to obtain [3]. Chest x-ray (CXR) findings of lobar consolidation is associated with bacterial pneumonia, while mild interstitial changes or infiltratesare associated with viral pneumonia. Lobar consolation is considered to be a more specific outcome measure for pneumococcal pneumonia by the World Health Organization (WHO), leading the WHO to develop = a methodology to standardize the radiologic definitions of childhood pneumonia for use in pneumococcal vaccine efficacy trials and epidemiology studies [3]. Primary endpoint pneumonia (PEP), defined as the presence of consolidation or pleural effusion, has since been used as an endpoint in a number of vaccine efficacy and impact studies [4–8]. In addition to PEP, the WHO methodology has conclusions for other infiltrates, normal (i.e. no consolidation, other infiltrates, or effusion), and uninterpretable.

Although the use of radiological endpoints is valuable for assessing vaccine efficacy and impact, it requires radiologist or physician engagement that is time- and cost-intensive. Most evaluations of PCV impact on radiological pneumonia have not used the WHO methodology for CXR interpretation [9]. Among studies that have used the WHO methodology, the number of CXR images analyzed varies given different resource constraints. Smaller studies have evaluated approximately 4,000 CXRs in the 3 years prior to and post-PCV introduction [10], while larger time series analyses have evaluated over 72,000 hospitalizations over 14 years (~10,000 PEP cases) [11], and 2.7 million visits over 9 years (~13,000 PEP cases) [8]. Automating the CXR reading task could not only standardize CXR interpretations across time and settings, but could also reduce the resources required to conduct these studies.

In addition to resource constraints, the subjective nature of the reading process and the varying level of expertise among radiologists or physicians can lead to considerable inter and intra-observer variability [12]. The WHO methodology is designed to standardize interpretations of CXRs, which is important for accurately determining the impact of PCVs in clinical trials and observational studies. In a randomized controlled trial of the 7-valent PCV, per protocol vaccine efficacy against radiological pneumonia, read by a radiologist at the point of care, was 20.5% in children less than 5 years of age [4]. After reevaluating the CXRs using the WHO methodology, vaccine efficacy against radiological pneumonia increased to 30.3% due to the improved specificity of the endpoint [13]. This illustrates the potential for discordant interpretations by humans and the impact it can have on evaluating interventions as well as prevalence estimates and epidemiological trends in disease.

Recent advancements in deep learning have enabled the automation of CXR reading at a performance comparable to experienced radiologists [14–17]. Automating the CXR reading task may improve sample efficiency by reducing discrepancies in interpretation and facilitating more widespread application of radiological endpoints in epidemiological research. In one previous study by Mahomed et al., the researchers automated the recognition of primary-endpoint pneumonia (PEP) using lung segmentation and texture analysis [18], using images from the same dataset as our current study. The analysis was run on data from the South African research site, which is only one of 7 sites in the PERCH (Pneumonia Etiology Research for

Child Health) study. The study achieved an area under the receiver operator characteristic curve (ROC) of 0.85 (95%CI: 0.823–0.876) for PEP within the South African dataset. In this study, we were able to train the model on the entire dataset from all research sites. We opted to use the deep learning approach, which requires less feature engineering than classical image analysis that typically involves manual filter selection. The final model will be tested on additional external datasets to further validate its performance, and to better understand its limitations beyond the PERCH study.

## Methods

The WHO methodology classifies pediatric CXRs into four endpoint conclusions: 'PEP' (primary endpoint pneumonia, i.e. consolidation or pleural effusion), 'other (non-endpoint) infiltrates', both 'PEP and other infiltrates', or 'Normal' (no consolidation, infiltrate, or effusion). The two PEP categoroies were merged into one in the analysis to represent "any PEP". To train a deep learning algorithm for this classification task, we utilized transfer learning where we pretrained a model on CheXpert, a large public CXR dataset [16], and finetuned it on the smaller Pneumonia Etiology Research for Child Health (PERCH) dataset with CXRs labeled according to the WHO methodology [12], and then tested it on two pediatric CXR datasets released by the WHO [3, 19].

### CheXpert dataset

The CheXpert dataset consists of 224,316 CXRs from 65,240 patients seen at Stanford Hospital inpatient and outpatient centers between October 2002 and July 2017 [16] CheXpert is primarily an adult CXR dataset containing 224,313 images from adults and 3 images from newborns. Natural language processing was used to extract text from radiology reports and label images as positive, negative, and uncertain for the presence of 14 common chest radiographic observations (S1 Fig). Further details can be found in Irvin, J. et al. [16].

### PERCH dataset

The PERCH study was a seven-country case-control study of causes and risk factors of childhood pneumonia in Africa and Asia [20, 21]. Cases were children between 1–59 months of age who were hospitalized with WHO-defined (pre-2013 definition) severe or very severe pneumonia [22, 23]. A total of 4,172 CXR images were available from 4,232 cases enrolled between August 2011 and January 2014. The PERCH study protocol was approved by the Institutional Review Boards or Ethical Review Committees for each of the seven institutions and at The Johns Hopkins School of Public Health. Parents or guardians of participants provided written informed consent, and all data were fully anonymized [24].

PERCH images were labeled by a 14-person reading panel comprised of radiologists and pediatricians from 7 study sites, along with a 4-person arbitration panel, consisting of radiologists experienced with the WHO methodology. Each image was reviewed by two randomly selected reviewers; images that received a discordant interpretation were then reviewed by two randomly selected arbitrators who were blinded to the previous interpretation. Discordant interpretations during arbitration were resolved through a final consensus discussion. Images were classified as PEP, other infiltrates, both PEP and other infiltrates, normal or uninterpretable (Table 1). Further details can be found in Fancourt et al. [12, 25].

**Table 1. Conclusions of CXR-reading by radiologists and pediatricians in training (PERCH) and testing (WHO) datasets.**

| Image Class | Training Dataset: PERCH (N = 4172) | | | | | Test Dataset: WHO (N = 431) | |
|---|---|---|---|---|---|---|---|
| | Final Conclusion (N = 4,172) | Round-1 Conclusions by Primary Readers | | Round-2 Conclusions by Arbitrators | | WHO-Original (n = 222) | WHO-CRES (n = 209) |
| | | (N = 4,172) | | (n = 2,358) | | | |
| | | Concordant (n = 1,814) | Discordant (n = 2,358) | Concordant (n = 1,144) | Discordant (n = 1,214) | | |
| Primary Endpoint Pneumonia | 1,075 (25.8%) | 458(11%) | 617(14.8%) | 228(9.7%) | 389(32%) | 90 (40.5%) | 71(34.0%) |
| Other Infiltrates | 993 (23.8%) | 361(8.7%) | 632(15.1%) | 276(11.7%) | 356(29.3%) | 44 (19.8%) | 26 (12.4%) |
| Normal | 1,692 (40.6%) | 854(20.5%) | 838(20.1%) | 521(22.9%) | 317(26.1%) | 75 (33.8%) | 106 (50.7%) |
| Uninterpretable | 412 (9.8%) | 141(3.4%) | 271(6.4%) | 119(5%) | 152(12.5%) | 13 (5.9%) | 6 (2.9%) |

## WHO-original and WHO-CRES datasets

In 1997, WHO released a teaching dataset with 222 CXRs to support the standardized interpretation of radiological pneumonia in children [3]. Each image was read by 20 radiologists and clinicians and labeled as PEP, other infiltrates, normal or uninterpretable. In the released dataset, 124 images were labeled as high agreement images since more than two-thirds of readers agreed on a single conclusion for these images. We refer to the remaining images in the dataset as low agreement images.

Two decades later, the WHO initiated the Chest Radiography in Epidemiological Studies (CRES) project to further clarify their classification methodology, with the objective of improving inter-observer agreement for each of the 3 endpoints [19]. The published WHO-CRES dataset contains only high agreement images (N = 209). Of these images, 176 were contributed by PERCH, including 14 uninterpretable images.

## Training procedures

We first pretrained the model on CheXpert to classify CXRs as positive or negative for 14 radiological findings with ImageNet initialization. The model was then finetuned on PERCH to detect PEP, other infiltrates, and normal findings with CheXpert initialization (S1 Fig). Initializing the model training with pretrained weights allows the model to achieve high performance on a smaller dataset by leveraging knowledge from models trained on a larger dataset [26]. In several previous studies using CXR images, DenseNet121 was selected as the convolutional neural network architecture [15–17]. Following the approach of previous studies, we also tried multiple available network architectures that have shown top-ranked performance on image classification including ResNet50 [27], InceptionV3 [28],VGG16 [29], NASNetMobile [30], NASNetLarge, Xception [31], DenseNet121 [32], and InceptionResnetV2 [33]. The DenseNet121 and NASNetLarge produced the highest overall AUC scores across the 3 WHO-defined categories, but DenseNet121 had slightly better performance on PEP and its relatively smaller size posed less risk of overfitting on a small dataset. Therefore, DenseNet121 was the chosen architecture in this study.

For pretraining on CheXpert, we follow the same process used in Irvin, J. et al. [16], where images were downscaled to $320 \times 320$ pixels, normalized based on the ImageNet training set, and augmented with a 50% random horizontal flipping and affine transformation such as rotate, shear, and translate by 10 degrees or 10%. Adam optimizer with default $\beta$ parameters ($\beta_1 = 0.9, \beta_2 = 0.999$) was used. We fixed the learning rate at $1 \times 10^{-4}$ throughout the training with a batch size of 16 images. Class imbalance was handled by reweighting the binary cross

entropy losses of each class by its inverse class frequency. The models were trained for 3 epochs with model checkpoint being saved and evaluated on the default validation set after every epoch. In the CheXpert study, the researchers tried multiple approaches to handle the uncertain labels. We opted for simple binary mapping and coded all "uncertain" labels with 0, noting that variations in coding for uncertain labels during pretraining had minimal impact on the transfer learning results.

Prior to finetuning on PERCH, images were manually cropped to focus on the pulmonary area and exclude body parts such as the abdomen, head, legs and arms (S2 Fig), in order to prevent the model from being adversely impacted by learning irrelevant features [34]. The manual cropping process also helps generate masks to train an image segmentation model to automate the cropping in the future (S3 Fig). The model was initiated with the weights from the best pretrained model on CheXpert. We used 10-fold cross-validation to reduce potential bias in model evaluation. The dataset was split into ten non-overlapping sets of images, and trained for 10 times with each set being held as a validation set each time. The area under receiver operating characteristic curves (AUCs) were calculated from the validation set and averaged across the 10 folds. The 95% confidence interval of the AUC was calculated using the non-parametric DeLong method [35].

During fine-tuning, we reduced the image size to $224 \times 224$ pixels, which yielded slightly better performance than $320 \times 320$, and kept the same augmentation as in the pre-training. We trained the networks with a batch size of 32 and used an initial learning rate of $1 \times 10^{-4}$, which was reduced by a factor of 10 each time the loss plateaued on the validation set. Early stopping was performed by saving the model after every epoch and choosing the saved model with the lowest validation loss. Although freezing lower layers in transfer learning has previously yielded better results [36], we found that freezing any of the lower layers resulted in suboptimal results compared to updating the entire model.

For both pretraining and fine-tuning, the parameters of the network were initialized with parameters from the pretrained network, except for the final fully connected layer, which was replaced with a new fully connected layer producing an output with the same dimension as the number of outcome classes. The weight of the replace layer was initialized with Glorot/Xavier Uniform initializer, with bias terms being set to zero. The outputs were then activated using sigmoid function to produce predicted probabilities of the presence of each of the outcome classes.

## Model evaluation

Uninterpretable images were removed from all analyses. AUCs were calculated separately for high and low agreement WHO test set images. We also compared model confidence between high and low agreement images for each outcome, using predicted probabilities as a measure of model confidence.

In addition to the WHO test sets, We also evaluated the PERCH model on its own hold-out set. The hold-out set included concordant and discordant images, analogous to the high and low agreement images in the WHO datasets. To create the hold-out set, we selected 150 images (50 per class) with concordant interpretations at either the primary or arbitration readings and another 150 images with discordant interpretations during arbitration. The remaining 3,460 images were used for training.

Discordant images were the most difficult for pediatricians and radiologists to interpret and required a consensus discussion to assign the final conclusion. To investigate how the model classified hard-to-interpret images, we used the same 150 discordant PERCH images described above as a hold-out test set and retrained the model on the remaining images

(n = 3,610). Since these discordant images required arbitration, a total of 4 readings for each image (two reviewers and two arbitrators) were available. For each of the 4 readings, we computed its sensitivity (recall), specificity, and positive predictive value (precision) against the final conclusion. We plotted the operating points from each of these readings along with the model's receiver operating characteristic (ROC) curve and Precision-Recall (PR) space to compare model results to that of the readers.

We also trained a model on all PERCH images with concordant interpretations only at either the primary or arbitration readings (n = 2,698) and used it to predict conclusions for all images that received a discordant interpretation during the arbitration reading (n = 1,062). Given the disagreement between highly trained arbitrators, the certainty of conclusion for these images is assumed to be relatively low. It is of interest to see what conclusions neural networks would assign to these hard-to-interpret images after being trained on images for which the certainty of conclusion is relatively high.

We assessed whether the model could correctly highlight the diseased area on a CXR image using guided Gradient-weighted Class Activation Mappings (Grad-CAMs). Grad-CAMs produce both a low resolution highlight (i.e. heat map) of the regions important to a class prediction and a high resolution class-discriminative visualization [37].

An small ablation experiment was conducted to evaluate the extent to which model performance was impacted by image cropping. We trained the model on uncropped images from the PERCH dataset and tested it on the WHO datasets.

## Results

Table 2 shows the validation and test AUCs achieved by the PERCH model on the PERCH validation, PERCH test, and WHO test datasets. The validation AUCs were 0.928 for PEP, 0.780 for other infiltrates and 0.897 for normal. The model achieved better performance on the external test set (WHO-Original and WHO-CRES images); the test AUCs increased to 0.977 for PEP, 0.891 for other infiltrates and 0.951 for normal.

Further analysis showed that the increase in model performance was related to the larger proportion of (78.78%) of high-agreement images in the WHO dataset. The PERCH model predicted PEP almost perfectly on WHO high agreement images (AUC = 0.993 and 0.996 for WHO-Original and WHO-CRES images respectively) but performance dropped by 14% on WHO low agreement images (AUC = 0.845 for WHO-Original images) (Table 2). A similar decline in performance was observed on discordant images in the PERCH dataset. The

**Table 2. AUROC scores (averaged across 10-fold) on the validation set, and WHO test set by level of inter-observer agreement of the image labels.**

| Category | Validation Results | Test Results | | | | | | | |
|---|---|---|---|---|---|---|---|---|---|
| | PERCH* (n = 346) | PERCH | | WHO All (N = 410) | WHO-Original | | | WHO-CRES |
| | | Concordant (n = 150) | Discordant (n = 150) | | High (n = 120) | Low (n = 88) | High + Low (n = 208) | High (n = 203) |
| Primary Endpoint Pneumonia | 0.928 (0.919,0.938) | 0.944 (0.930,0.957) | 0.859 (0.837,0.879) | 0.977 (0.974,0.981) | 0.993 (0.990,0.996) | 0.845 (0.817,0.873) | 0.952 (0.943,0.960) | 0.996 (0.995,0.998) |
| Other Infiltrates | 0.780 (0.764,0.797) | 0.810 (0.788,0.832) | 0.741 (0.715,0766) | 0.891 (0.879,0.903) | 0.969 (0.957,0.980) | 0.726 (0.692,0.759) | 0.856 (0.838,0.875) | 0.935 (0.919,0.950) |
| Normal | 0.897 (0.887,0.907) | 0.896 (0.880,0.911) | 0.788 (0.765,0.812) | 0.951 (0.945,0.957) | 0.995 (0.992,0.997) | 0.749 (0.714,0.784) | 0.921 (0.909,0.932) | 0.974 (0.968,0.980) |

* Average sample size of the 10-fold validation set.

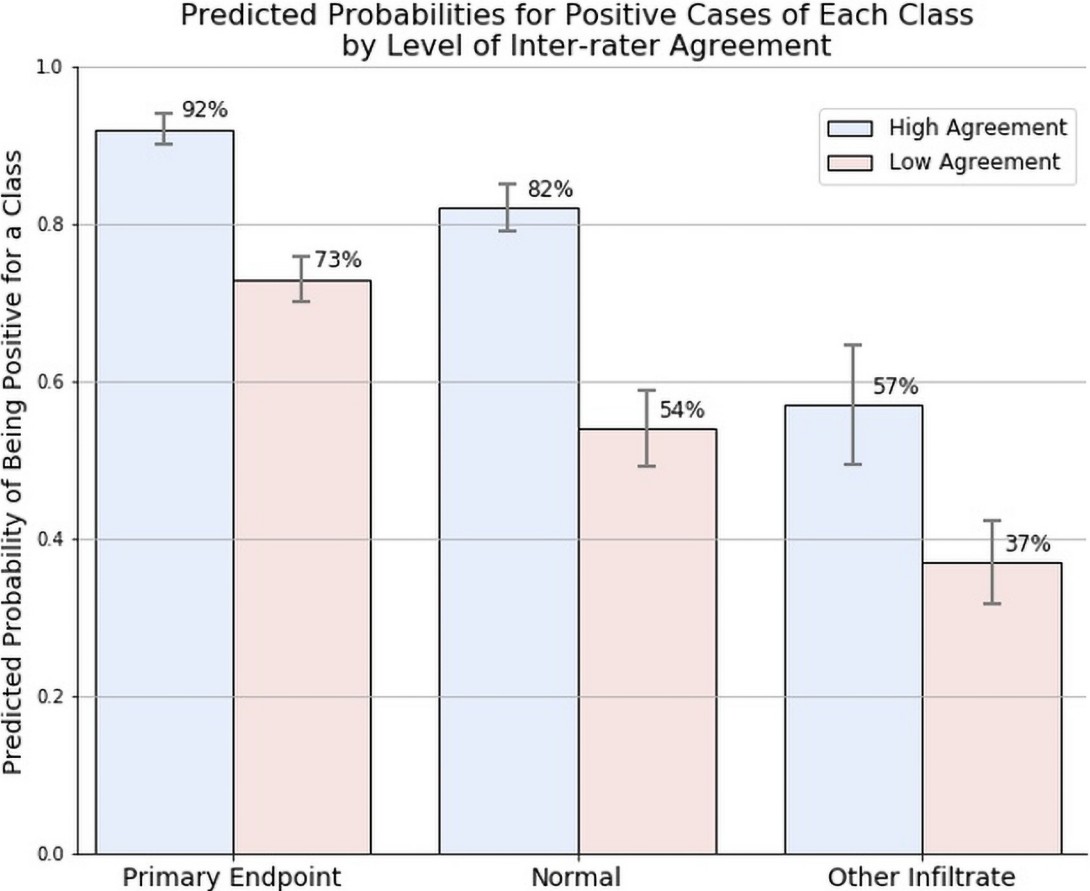

**Fig 1. Comparison of model predicted probabilities and 95% confidence intervals by endpoint and level of human reader agreement.**

predicted probabilities of classification were higher for high versus low agreement images across all classes, with higher predicted probabilities for PEP compared to other classes (Fig 1).

To illustrate the model performance in the context of discontinuous accuracy scores, we took the optimal cut-off of AUC score that maximizes the Youden's index [38]. Among high-agreement WHO images, the test AUCs correspond to 95.3%, 96.7%, 91.1% of sensitivity, and 95.5%, 96.8%, 91.8% of specificity for PEP, normal and other-infilrates, respectively. Among low-agreement WHO images, the sensitivity dropped to 76.7%, 69.7%, 71.9% and specificity dropped to 76.9%, 70%, 66.4% for the three outcomes, respectively. The specificity is higher than sensitivity for all outcomes, reflecting the intended specificity of the WHO definition, a criteria that is important for estimating vaccine efficacy and impact.

Fig 2 shows the comparison between the model's prediction and the 4 readings against the final conclusion determined during the consensus discussion for discordant images. ROC and precision-recall (PR)-curves from 10-fold validation are displayed with the average curves highlighted in blue. Four operating points are also displayed representing the conclusions given by the 4 readings. Averaging across ten folds, the model's performance on all three classes was better than the 4 readings provided by pediatricians and radiologists.

Fig 3a and 3b show a side by side comparison of WHO's annotation and the localized regions that the model predicted to be most indicative of an outcome class. The model

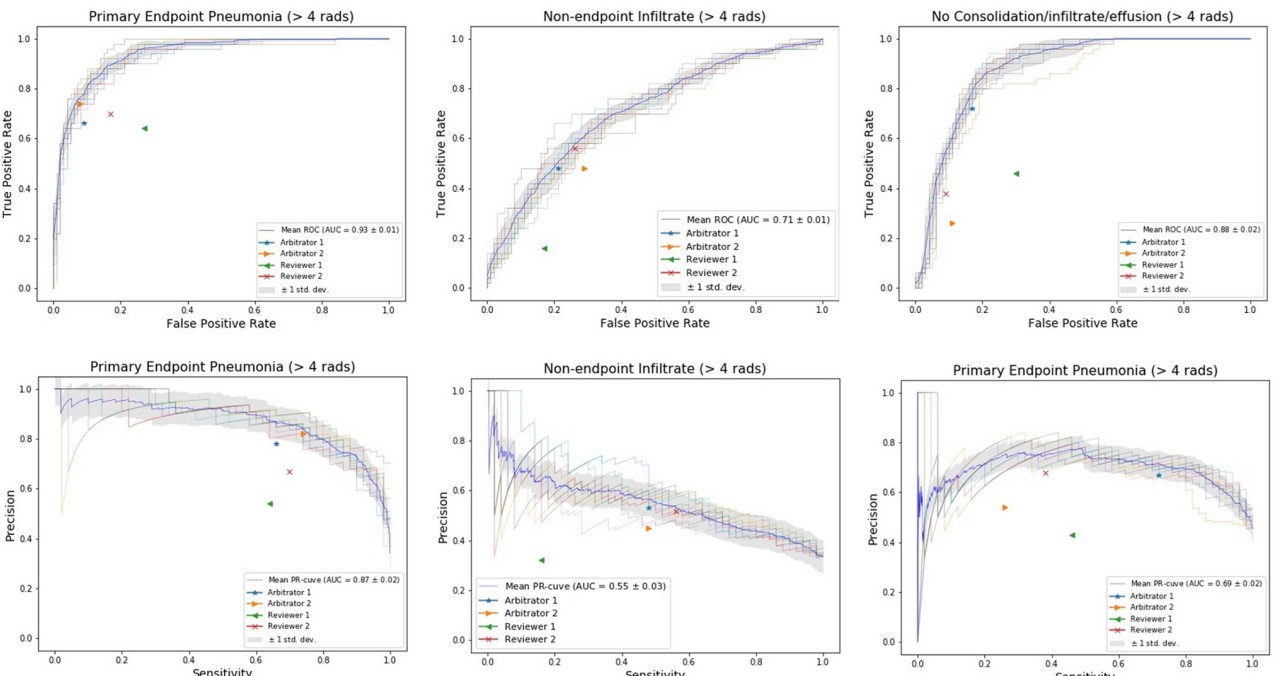

**Fig 2. Comparison of model performance to radiologist and pediatricians on discordant images.** The four operating points represent the conclusions given by the 4 readings. The lines represent model's performance, with the average of 10-fold validation in blue color. The top rows shows the receiver operating characteristic (ROC) curve and the bottom shows the Precision-Recall (PR) curve.

identified the area of infection for PEP with high accuracy (p = 0.980). The identification of other infiltrates was relatively less accurate given the dispersed nature of the infected area (p = 0.917).

## Additional results

The ablation experiment showed that image cropping improved model performance by 1–3% on AUC (S1 Table).

Table 3 shows the agreement between the model's prediction and the final conclusion assigned to images that received a discordant interpretation during arbitration. The model's prediction agreed with that of pediatricians and radiologists on 60% of the discordant images (Table 3). Agreement was highest for PEP and lowest for other infiltrates. Predicted probabilities were higher when the model's conclusion agreed with the final conclusion for PEP (85.6% vs 70%, p<0.001) and normal (76.9% vs 69.9%, p<0.001), but not for other infiltrates (60.6% vs 60.5%). No association was found between agreement status and other variables.

In S2 Table, we present various comparison matrices, where diagonal cells indicate agreement between the model and the final conclusion, and off diagonal cells indicate disagreement. The model and the readers were both more likely to classify normal images as other infiltrates than PEP. However, the model was more likely to classify other infiltrates as normal than PEP, while readers were more likely to classify other infiltrates as PEP than normal. When the model's prediction differed from the final conclusion, 22% of its predictions agreed with the conclusion that received a majority of votes where one reviewer agrees with one arbitrator, and the other reviewer disagrees with the other arbitrator prior to final consensus discussion (S4 Fig).

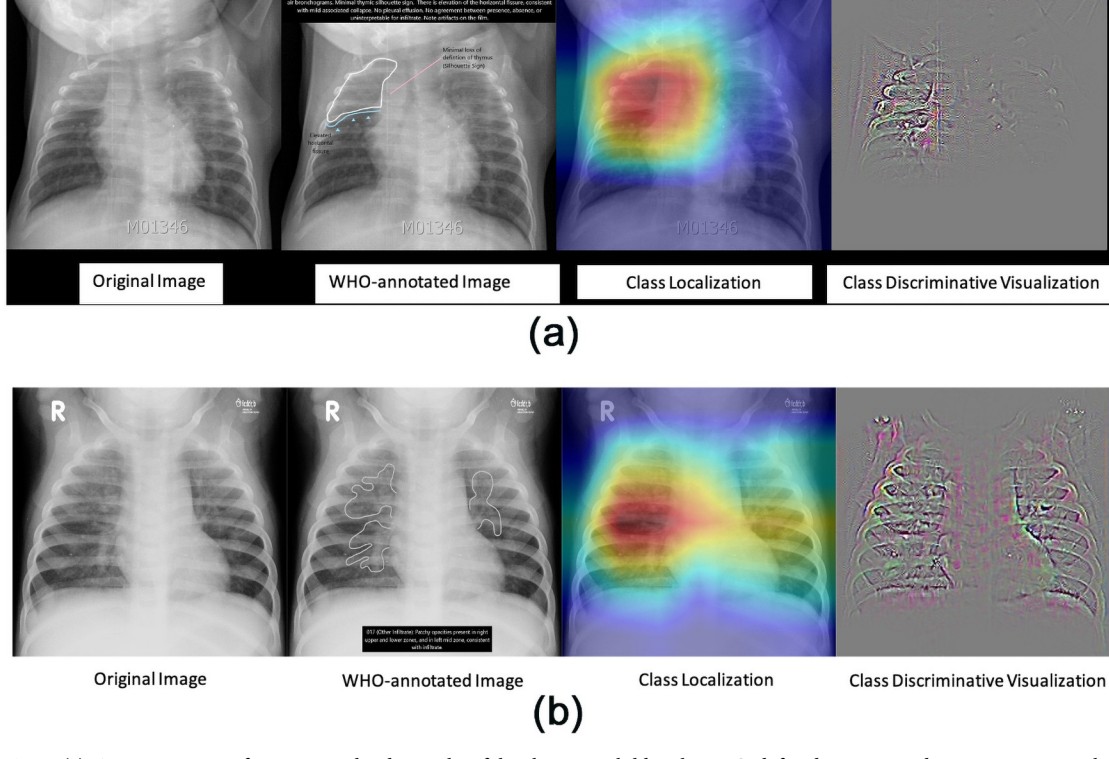

**Fig 3.** (a). Activation map of PEP. Frontal radiographs of the chest in a child with WHO-defined primary endpoint pneumonia; the child is rotated to the right with dense opacity in the right upper lobe; the model localizes consolidation with a predicted probability $p = 0.980$; the discriminative visualization shows fine-grained features important to the predicted class. (b). Activation map of other-infiltrates. Frontal radiograph of the chest presents patchy opacity consistent with non-endpoint infiltrate. The model correctly classifies the image as infiltrate with a probability of $p = 0.917$ and localizes the areas of opacity. The class discriminative visualization highlights important class features.

## Discussion

This is one of the first studies to automate detection of WHO-defined radiological pneumonia using deep learning. The neural network's performance is better than a previous study using classical image texture analysis on a subset of PERCH images. The increase in performance could be due to either the modeling approach, or the increase in sample size. The model's performance is also comparable to the performance of human readers on all 3 WHO-defined pneumonia categories. We also noticed a boost in model performance on external test sets, and were able to identify the root cause to be related to inter-rater agreement in the human-assigned image labels from the external test sets.

Model performance was higher on the WHO test images than on the validation or test images from PERCH. This is atypical, as one would expect an independent test sample to have poorer classification performance than validation or test samples retrieved from the training set. We concluded that the improvement in AUCs on the WHO datasets, ranging from 5–11% across the 3 conclusions, was due to a larger proportion of images having high inter-observer agreement. This also explains why model performance was higher on PERCH concordant than discordant images.

Previous studies have shown that inter-observer agreement is highest for PEP and lowest for other infiltrates, even when a rigorous standardization process is implemented [3, 12]. This

**Table 3. Final conclusion and model prediction on discordant CXR images (N = 1,062) by key features.**

| | Model = Final conclusion (n = 637) | Model ≠ Final conclusion (n = 425) |
|---|---|---|
| Final conclusion (n,%) * | | |
| Primary Endpoint Pneumonia (PEP) | 257(40.4%) | 132(31.1%) |
| Other Infiltrates (OI) | 149(23.4%) | 207(48.7%) |
| Normal | 231(36.3%) | 86(20.2%) |
| CXR + (PEP or OI)** | 558/789(70.7%) | 187/273(68.5%) |
| Predicted Probability (mean, SD) | | |
| PEP * | 85.6%(0.16) | 70.0%(0.20) |
| OI | 60.6%(0.13) | 60.5%(0.12) |
| Normal* | 76.9%(0.15) | 69.9%(0.16) |
| Gender (n,%) | | |
| Male | 351(55.1%) | 233(54.8%) |
| Female | 286(44.9%) | 192(45.2%) |
| Age in months (mean, SD) | 10.50(10.69) | 11.19(10.96) |
| Countries (n,%) | | |
| Bangladesh | 70(11.0%) | 52(12.2%) |
| Gambia | 93(14.6%) | 70(16.5%) |
| Kenya | 82(12.9%) | 68(16.0%) |
| Mali | 81(12.7%) | 48(11.3%) |
| South Africa | 186(29.2%) | 105(24.7%) |
| Thailand | 33(5.2%) | 27(6.4%) |
| Zambia | 92(14.4%) | 55(12.9%) |

* $p < 0.0001$ for Pearson's chi-squared test or two-proportion z-test.

** Differences between PEP and OI are ignored so a greater number of images have concordant results.

also appears to be the case in our study, where the model achieved the best classification performance for PEP. In contrast, a finding of "other infiltrates" is known to have the lowest inter-observer agreement among human readers and was also the most difficult for the model to identify. Compared to PEP, the model generated lower predicted probabilities for classifying other infiltrates even when its conclusions agreed with the readers.

We also observed that the model was more likely to classify images as normal than PEP when it disagreed with the conclusion of other infiltrates, while the readers were more likely to classify these images as PEP. It is not clear if the observed difference reflects any underlying bias in the labeling procedure or in the model's prediction. However, this may lower the potential of the model to overestimate vaccine impact or pneumonia disease burden when applied to new settings.

To better understand the predictive qualities of the model, two radiologists reviewed a selection of the WHO images that were misclassified as PEP by the model. For PEP images that the model falsely predicted as negative, the CXR films were generally more penetrated, with more photons hitting the film. The areas of consolidation that were missed also tended to be adjacent to the cardiac silhouette or scapula. One of the false negatives may be a mislabeled image as neither radiologist could identify any consolidation. On images with false positive predictions, CXR films appeared to be less penetrated. This is consistent with human intuition, as under-penetration may obscure lung parenchyma, which happens more focally with consolidation. This underscores the importance of quality control, including technical processes and patient positioning, in research studies using pediatric CXRs.

## Limitations

One potential limitation of training a large neural network on a small dataset is overfitting. The validation loss on average plateaued after 3–5 epochs of fine-tuning. However, transfer learning allows the model to converge faster without excessive training [26]. Although the model achieved high performance on the WHO dataset, additional validation may be needed when applying it to new settings given that the WHO dataset is a teaching set with mostly high agreement images. The model is developed for epidemiologic purposes, and not intended to be used as a diagnostic tool in clinical practice. The model is trained on images from children hospitalized for WHO-defined severe and very severe pneumonia and its application should be restricted to images from children with a diagnosis of pneumonia to avoid misclassification of non-pneumonia cases as PEP.

## Conclusion

A deep learning model can identify primary endpoint pneumonia in children at a performance comparable to human readers. It can be implemented without manual feature engineering and achieves better performance than classical image analysis. This study lays a strong foundation for the potential inclusion of computer-aided pediatric CXR readings in vaccine trials and epidemiology studies.

## Supporting information

**S1 Fig. Transfer learning illustration.**
(PNG)

**S2 Fig. Sample images and cropping.**
(PNG)

**S3 Fig. U-Net for auto-cropping.**
(PNG)

**S4 Fig. Flow of model prediction disagrees when model disagrees with final conclusion.**
(PNG)

**S1 Table. AUROC scores of models trained on raw PERCH images and tested on WHO.**
(DOCX)

**S2 Table. Matrix of model predictions by final and majority conclusions.**
(DOCX)

## Acknowledgments

We are grateful to Jeremy Irvin from Stanford Machine Learning Group for being helpful and responsive to the questions regarding the CheXpert dataset; we would also like to thank Antong Chen and Ilknur Icke from for their insights on the image analysis, Albert (Haigang) Liu for conducting preliminary analysis, Harvey (Zhezhong) Jiang and Kelly M. Cahill for cropping the CXR images, Cynthia Goldberg for reviewing the images and labels in question, and Di Ai for proofreading the manuscript. Last but not least, we would like to express our sincere gratitude to Christine Prosperi from Johns Hopkins University for her support and help with questions related to the PERCH dataset. We would also like to extend our thanks to the entire Pneumonia Etiology Research for Child Health (PERCH) Study Group for this opportunity of collaboration.

## Author Contributions

**Conceptualization:** Craig S. Roberts, Wanmei Ou, Maria Deloria Knoll.

**Data curation:** Nicholas Fancourt, Maria Deloria Knoll.

**Formal analysis:** Yiyun Chen.

**Funding acquisition:** Maria Deloria Knoll.

**Investigation:** Maria Deloria Knoll.

**Methodology:** Yiyun Chen, Gregory V. Goldmacher.

**Project administration:** Maria Deloria Knoll.

**Software:** Yiyun Chen.

**Supervision:** Nicholas Fancourt, Maria Deloria Knoll.

**Validation:** Gregory V. Goldmacher.

**Visualization:** Yiyun Chen.

**Writing – original draft:** Yiyun Chen.

**Writing – review & editing:** Yiyun Chen, Craig S. Roberts, Wanmei Ou, Tanaz Petigara, Maria Deloria Knoll.

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
