## [Decision Letter · Decision Letter 0]

29 Mar 2021

PONE-D-21-05523

Deep Learning for Classification of Pediatric Chest Radiographs by WHO's Standardized Methodology

PLOS ONE

Dear Dr. Chen,

Thank you for submitting your manuscript to PLOS ONE. After careful consideration, we feel that it has merit but does not fully meet PLOS ONE’s publication criteria as it currently stands. Therefore, we invite you to submit a revised version of the manuscript that addresses the points raised during the review process.

We look forward to receiving your revised manuscript.

Kind regards,

Khanh N.Q. Le

Academic Editor

PLOS ONE

Journal Requirements:

6. Thank you for providing the following Funding Statement: 

[Yiyun Chen, Tanaz Petigara, Wanmei Ou, Craig S. Robert, Gregory V. Goldmacher were employees of  Merck & Co., Inc. during the conduct of the study.

Maria Deloria Knoll received grant from Merck & Co. to cover expenses related to preparing the PERCH dataset shared for use in the study, and for consulting on the manuscripts.].   

We note that one or more of the authors is affiliated with the funding organization, indicating the funder may have had some role in the design, data collection, analysis or preparation of your manuscript for publication; in other words, the funder played an indirect role through the participation of the co-authors.

If the funding organization did not play a role in the study design, data collection and analysis, decision to publish, or preparation of the manuscript and only provided financial support in the form of authors' salaries and/or research materials, please review your statements relating to the author contributions, and ensure you have specifically and accurately indicated the role(s) that these authors had in your study in the Author Contributions section of the online submission form. Please make any necessary amendments directly within this section of the online submission form.  Please also update your Funding Statement to include the following statement: “The funder provided support in the form of salaries for authors [insert relevant initials], but did not have any additional role in the study design, data collection and analysis, decision to publish, or preparation of the manuscript. The specific roles of these authors are articulated in the ‘author contributions’ section.”

If the funding organization did have an additional role, please state and explain that role within your Funding Statement.

Please also provide an updated Competing Interests Statement declaring this commercial affiliation along with any other relevant declarations relating to employment, consultancy, patents, products in development, or marketed products, etc.  

Reviewers' comments:

Reviewer's Responses to Questions

**Comments to the Author**

1. Is the manuscript technically sound, and do the data support the conclusions?

Reviewer #1: Partly

Reviewer #2: Yes

Reviewer #3: Yes

Reviewer #4: Yes

2. Has the statistical analysis been performed appropriately and rigorously? 

Reviewer #1: Yes

Reviewer #2: Yes

Reviewer #3: Yes

Reviewer #4: Yes

3. Have the authors made all data underlying the findings in their manuscript fully available?

Reviewer #1: No

Reviewer #2: Yes

Reviewer #3: Yes

Reviewer #4: No

4. Is the manuscript presented in an intelligible fashion and written in standard English?

Reviewer #1: Yes

Reviewer #2: Yes

Reviewer #3: Yes

Reviewer #4: Yes

5. Review Comments to the Author

Reviewer #1: The authors have used a DenseNet-121 model pretrained on the CheXpert data and fine-tuned it to predict CXRs as showing PEP, other infiltrates, or normal lungs. They have experimented with in-house and WHO test sets. They conclude that the model delivers superior performance with concordant images from the WHO test set compared to discordant images in the PERCH test set.

What is the objective of the study? Are the authors trying to propose a novel architecture that can generalize to real-world PEP detection? The authors mention “The model is trained on images from children hospitalized for WHO-defined severe and very severe pneumonia and its application should be restricted to images from children with a diagnosis of pneumonia to avoid misclassification of non-pneumonia cases as PEP.” What is the purpose of using a trained model to classify a CXR as showing PEP when the patient is already diagnosed to have PEP? The model indeed needs to function as triage in resource-constrained settings.

What is the effect of increasing/decreasing spatial resolution in the proposed approach? This reviewer is not clear why the authors resized the images to 320 x 320 pixels during the pretraining stage and then 224 x 224 in the fine-tuning stage. A detailed discussion on using different spatial resolutions and their impact on model performance shall be provided. A statistical significance analysis is required in this regard.

This reviewer is unclear why the authors worked with only eight ImageNet models. What is the performance obtained by a baseline, sequential CNN? Deeper models are not always better for medical image analysis tasks. There is no comprehensive analysis provided while pretraining and finetuning with different models. How did the authors optimize the architecture and hyperparameters of these models? Did they truncate these models at an optimal layer to suit the classification problem under study? These details are not discussed. Also, the authors didn’t discuss how they managed the class-imbalance problem during the pretraining stage.

At present there are several studies available that discuss CXR modality-specific pretraining in the context of classifying lung diseases, a few are mention below:

https://ieeexplore.ieee.org/stamp/stamp.jsp?arnumber=8621525

https://pubmed.ncbi.nlm.nih.gov/33180877/

The authors need to perform a comprehensive literature survey in this regard and cite these studies. This reviewer does not think reference [27] suits here and should be replaced with some of the aforementioned references.

This reviewer is not convinced about the AUC curves obtained with various test sets. The default classification threshold used by these models is 0.5. What are the sensitivity and specificity obtained at varying operating thresholds? How did the authors make a selection of the sensitivity and specificity at an optimum threshold that offers a good trade-off?

This reviewer is not convinced with the manual cropping of the lung regions from the CXR images. It is recommended to automate the process using U-Nets or other image segmentation models as that discussed in the SOTA like https://pubmed.ncbi.nlm.nih.gov/33180877/ and discuss the model performance in terms of segmentation evaluation metrics and see if the differences are statistically significantly different.

The authors mention in the discussion that “Results demonstrate that the deep learning approach can achieve better classification performance for PEP (by about 8%) than image texture analysis with no need for feature engineering. This is not agreeable. They have not performed any kind of handcrafted texture analysis in this study. The literature might not have used the same train and test sets that the authors propose in this study. It is recommended to perform a texture analysis with varying texture descriptors and then discuss how the performance compares to the deep learning methods.

The authors mention in the conclusion that “It can be implemented without manual feature engineering and achieves better performance than classical image analysis”. This is not agreeable for the reason aforementioned.

The authors shall mention the computational resources and deep learning frameworks used in this study.

Reviewer #2: It is a common practice to use transfer learning in image analysis. The paper presents a detailed methodology in classifying radiographs using transfer learning. Please see the following comments:

1. It would be nice if the authors could provide more technical details in pre-training using ImageNet initialization. This is because the number of classes is 1000, however, the pre-training only requires 14. If this is true, does that mean the weights in the final linear projection layer are re-initialized, instead of using the ImageNet initializations?

2. Similar to the point above, the S1 figure should have more details to precisely describe the pretraining process.

3. It is understandable that the focus of the paper is to present a practical way in radiograph classification using transfer learning. It would be nice if the authors could provide some future work.

Reviewer #3: Some Recommendations:

Highlight the originality of the paper as model that have been used is already available.

It is not clear whether cropping of CXRs is done manually or automated. Specify it.(line no 157)

Discuss the deep-learning model that is used in the study.

Also add the results obtained on various deep-learning models that you have compared.

Reviewer #4: The manuscript is good overall. This study based on deep learning method lays a strong foundation for the potential inclusion of computer-aided pediatric CXR readings in vaccine trials and epidemiology studies. There are some spelling mistakes need to be corrected，such as "trainig" in line 140.

6. PLOS authors have the option to publish the peer review history of their article (what does this mean?). If published, this will include your full peer review and any attached files.

Reviewer #1: No

Reviewer #2: No

Reviewer #3: **Yes: **Dr. Rahul Hooda

Reviewer #4: **Yes: **Rongguo Zhang

---

## [Author Response · Author response to Decision Letter 0]

12 May 2021

Reviewer #1: The authors have used a DenseNet-121 model pretrained on the CheXpert data and fine-tuned it to predict CXRs as showing PEP, other infiltrates, or normal lungs. They have experimented with in-house and WHO test sets. They conclude that the model delivers superior performance with concordant images from the WHO test set compared to discordant images in the PERCH test set.

Response: We would like to thank the reviewer for taking the time to give the paper a careful read. 

Reviewer #1: What is the objective of the study? Are the authors trying to propose a novel architecture that can generalize to real-world PEP detection? The authors mention “The model is trained on images from children hospitalized for WHO-defined severe and very severe pneumonia and its application should be restricted to images from children with a diagnosis of pneumonia to avoid misclassification of non-pneumonia cases as PEP.” What is the purpose of using a trained model to classify a CXR as showing PEP when the patient is already diagnosed to have PEP? The model indeed needs to function as triage in resource-constrained settings.

Response: 

WHO-defined clinical pneumonia is a general clinical definition of symptoms (i.e. cough, fever etc.) which does not indicate the etiology of the infection (i.e. bacterial vs viral infection). To support classification as an endpoint in vaccine efficacy and epidemiology studies the WHO convened an expert panel to define specific chest x-ray (CXR) findings specifically associated with bacterial pneumonia in children=. PEP, per this definition, is a specific type of pneumonia, more likely to be bacterial and with high relevance to pneumococcal infection. The major characteristic of PEP is the manifestation of consolidation on chest x-ray images. Consolidation refers to the alveolar airspaces being filled with fluid (exudate/transudate/blood), cells (inflammatory), tissue, or other material. In practice, implementation of PEP in bacterial pneumonia research studies has often involved a team of radiologists, with multiple readings per child and a consensus process to interpret the findings. By using the data and images of such a study, (e.g., PERCH), we are training the computer to provide this specific reading, which can enable greater scalability and consistency to these types of studies without the teams of radiologists to adjudicate.

We did not describe this background information in too much detail given concerns of conciseness. We have cited all the relevant studies that can provide this background knowledge for readers who are interested. Meanwhile to address the reviewer’s comment, we have now added between line 51-53, a description of consolidation and mild interstitial changes to clarify where the concept of PEP came from. We have also added more details to help establish the connection between pneumococcal pneumonia, bacterial infection, and lobar consolidation (i.e. PEP). Lastly, we reorganized the introduction in order to better delineate the rationale behind the development of a deep learning model for PEP. The ultimate objective of the study is to automate CXR reading for PEP to incorporate in large vaccine efficacy and epidemiology studies. 

Reviewer #1: What is the effect of increasing/decreasing spatial resolution in the proposed approach? This reviewer is not clear why the authors resized the images to 320 x 320 pixels during the pretraining stage and then 224 x 224 in the fine-tuning stage. A detailed discussion on using different spatial resolutions and their impact on model performance shall be provided. A statistical significance analysis is required in this regard.

Response: We have now added a description of how we select the hyperparameters under the training procedures section. Image size or “spatial resolution”, is one of the many hyperparameters that needs to be tuned during training. In machine learning/deep learning, hyperparameter tuning does not necessarily require a statistical significance test. Grid search or random search is typically used over a range of plausible values of different hyperparameters, and the combination that yields the lowest loss or highest accuracy will be selected for full model training. Example discussions and implementations can be found at: https://stats.stackexchange.com/questions/321805/statistical-significance-of-changing-a-hyperparameter

https://machinelearningmastery.com/hyperparameters-for-classification-machine-learning-algorithms/.

This can also been seen in the IEEE paper that the reviewer mentioned later in one of the comments (https://ieeexplore.ieee.org/stamp/stamp.jsp?arnumber=8621525). In that study, the researchers simply stated “The creators of ResNet recommend converting images to square sized images of either 224 by 224 pixels or 299 by 299 pixels before feeding them to ResNet. We opted for the higher resolution 299 by 299 for higher accuracy” as the rationale for choosing 299 pixels. 

In this study, the hyperparameters of interest include image size, learning rate, alpha and beta coefficient of adam optimizer, level of image augmentation, batch size, etc. Because the pretraining is a replication of the CheXpert study, we kept the same “special resolution” as the original study given that other researchers have already narrow down to the best resolution (320 by 320 pixels) for the proposed architecture. For finetuning on the PERCH dataset, we found 224 by 224 to yield a slightly higher AUC score than the 320 by 320 pixels. This detail has been added to the manuscript between line 180-181.

Reviewer #1: This reviewer is unclear why the authors worked with only eight ImageNet models. What is the performance obtained by a baseline, sequential CNN? 

Response: These 8 models cover most of the available architectures that have shown top notch results on image classification. Each of the eight model architectures have shown by previous studies to have significantly outperformed the previous generation of models. The eight models we tried were the same eight models applied by other researchers (https://arxiv.org/pdf/1901.07031.pdf), where the researchers found that DenseNet121 works best for chest radiograph images. Other studies on CXR images have also done similar analysis and found the DenseNet architecture to work better on CXR images than other available ones. As with the previous work, we found the DenseNet model to have high accuracy with training. This detail is provided between line 148-150 with all the relevant references. 

Reviewer #1: Deeper models are not always better for medical image analysis tasks. There is no comprehensive analysis provided while pretraining and finetuning with different models. How did the authors optimize the architecture and hyperparameters of these models? Did they truncate these models at an optimal layer to suit the classification problem under study? These details are not discussed. 

Response: We agree with the reviewer that deeper model is indeed not necessarily better, especially on a relatively smaller dataset. After finding the optimal architecture (i.e. DenseNet 121) for CXR images, by following the same procedure in previous studies, we had attempted freezing the weights at the lower layers of the model, in order to reduce the number of parameters need to be trained during back propagation. This has similar effect as training on a shallower model with smaller number of parameters. But it turned out that training on the entire model still yields better performance. We tried shallower models, such as VGG16, but its accuracy was over 10% lower than DenseNet121. We have added these details in the paper, along with a comparison to the IEEE article finding that the reviewer mentioned in one of the later comments between line 185-187. On the other hand, the pretraining procedure is usually not extensively described because it is typically a replication of previous studies, as we mentioned in the paper. Meanwhile, we have reorganized the paragraph to clarify that that the pre-training approach was a replication of the previous study from which we borrow the dataset. We have cited all the relevant papers in the manuscript for readers who are interested in more details. Meanwhile we also added more details on how we handled uncertain labels in the pretraining dataset between line 164-168, as this detail is unique to the current study, and is not directly available in the reference article. 

Reviewer #1: Also, the authors didn’t discuss how they managed the class-imbalance problem during the pretraining stage.

Response: Thank you for pointing this out and we have added a description of how the imbalance problem is handled on page between line 161-162.

Reviewer #1: At present there are several studies available that discuss CXR modality-specific pretraining in the context of classifying lung diseases, a few are mention below:

https://ieeexplore.ieee.org/stamp/stamp.jsp?arnumber=8621525

https://pubmed.ncbi.nlm.nih.gov/33180877/

The authors need to perform a comprehensive literature survey in this regard and cite these studies. This reviewer does not think reference [27] suits here and should be replaced with some of the aforementioned references.

Response: Reference 27 is a textbook cited often as a source of transfer learning technique, which as the reviewer has appropriately pointed out, may not be relevant here and hence we have removed it. We also agree with the reviewer that there are several studies, and in fact more than several papers available that used transfer learning on CXR images. A majority of these studies are about tuberculosis, along with a surge of COVID-19 deep learning papers over the past year. Transfer learning is almost a standard approach for deep learning in image classification, as reviewer 2 also mentioned in the opening comment. We did not cite all those studies as they are in different disease areas, none of these studies is the original study that proposed transfer learning in image classification, and none of these are the first to use transfer learning on CXR. Our goal is to focus the literature search on studies with direct relevance to the current study, including all the PCV vaccine disease burden studies using CXR finding as the outcome, along with deep learning method papers including the CheXpert paper (https://arxiv.org/abs/1901.07031), and its prior versions, the CheXNet paper (https://arxiv.org/pdf/1711.05225.pdf) and CheXNeXt paper (https://journals.plos.org/plosmedicine/article?id=10.1371/journal.pmed.1002686) from which we either borrowed the method or used the dataset. Meanwhile, we do see the value of citing the two papers that the reviewer mentioned that used CXR images from the perspective of methodology, so we have cited the aforementioned paper in the relevant part of the revised manuscript on line 186, and line 171, along with relevant description and comparison to the current study. 

Reviewer #1: This reviewer is not convinced about the AUC curves obtained with various test sets. The default classification threshold used by these models is 0.5. What are the sensitivity and specificity obtained at varying operating thresholds? How did the authors make a selection of the sensitivity and specificity at an optimum threshold that offers a good trade-off?

Response: We understand the reviewer’s desire to know the sensitivity and specificity and we have now included sensitivity and specificity in the results section. We did not report this piece of information originally because the goal of the study is not label generation, but model validation. Sensitivity and specificity are discontinuous arbitrary accuracy scores, whereas the AUC score is not used for forced choice but rather for assessing the pure predictive discrimination of a continuous prediction. It is also widely used in the literature for validating a deep learning model and has been the default metric used in previous literature that we have cited. However, we do understand that the concept of sensitivity and specificity is more relevant in a medical setting and is practical information when applied to the ultimate objective of the work, which is application to large scale research studies of childhood pneumonia. These metrics have now been added, along with relevant description in the manuscript between line 246-253.

Reviewer #1: This reviewer is not convinced with the manual cropping of the lung regions from the CXR images. It is recommended to automate the process using U-Nets or other image segmentation models as that discussed in the SOTA like https://pubmed.ncbi.nlm.nih.gov/33180877/ and discuss the model performance in terms of segmentation evaluation metrics and see if the differences are statistically significantly different.

Response: Automation is our goal as well, and indeed a U-Net can help to automate segmentation. However, the training of a U-Net is a supervised learning approach, and requires ground-truth bounding boxes. We are not aware of any open-source U-Net available to crop lung regions on pediatric CXRs, and this is why we manually cropped the images and created our own ground-truth masks or bounding box. We have already used the masks to train a U-Net to automatically identify the lung area, so in the future this process can be automated. But in the current study, given that we already have the ground truth bounding box, we do not see the need to apply the U-Net and regenerate the bounding box again. We have added a description of the U-Net in the supplementary material (see S3 Fig), along with a description between line 171-173 in the manuscript. We originally did not mention this detail because it is more relevant to future studies than the current one, but we are glad that the reviewer pointed this out, and we also see the value of adding it to the paper for future reference. 

Reviewer #1: The authors mention in the discussion that “Results demonstrate that the deep learning approach can achieve better classification performance for PEP (by about 8%) than image texture analysis with no need for feature engineering. This is not agreeable. They have not performed any kind of handcrafted texture analysis in this study. The literature might not have used the same train and test sets that the authors propose in this study. It is recommended to perform a texture analysis with varying texture descriptors and then discuss how the performance compares to the deep learning methods.

The authors mention in the conclusion that “It can be implemented without manual feature engineering and achieves better performance than classical image analysis”. This is not agreeable for the reason aforementioned.

Response: We agree with the reviewer that the original description is problematic because there are two potential reasons for the different results. The first is the different modeling approach, and the second is the difference in datasets. The reviewer has also raised a very good point on the population being different because in that study, the researcher only used data from some of the study sites, instead of all sites. We have revised the description between line 285-288, and also incorporated the reviewer’s point between line 87-96 in the introduction. Yet, it is not clear what the reviewer means by texture analysis in the context of deep learning. The difference between deep learning and classical image analysis is that “texture” no longer needs to be selected by the human user, and instead of texture descriptor, the convolutional filters is part of the trained parameters and network will decide for human what kind of “texture” that needs to be learned at each layer. It seems that the type of analysis that the reviewer is potentially implying may be beyond the scope of the current study. 

Reviewer #1: The authors shall mention the computational resources and deep learning frameworks used in this study.

Response: We have added this piece of information in the supplementary material. We will also be sharing the codes and trained model on github later.

Reviewer #2: It is a common practice to use transfer learning in image analysis. The paper presents a detailed methodology in classifying radiographs using transfer learning. Please see the following comments:

Response: We would like to thank the reviewer for taking the time to read the paper.

1. It would be nice if the authors could provide more technical details in pre-training using ImageNet initialization. This is because the number of classes is 1000, however, the pre-training only requires 14. If this is true, does that mean the weights in the final linear projection layer are re-initialized, instead of using the ImageNet initializations?

Response: We have added a paragraph to describe the weight initialization in more detail between line 188-194.

2. Similar to the point above, the S1 figure should have more details to precisely describe the pretraining process.

Response: We have added some details in the graph to indicate that the weight of the final layer was randomly initialized.

3. It is understandable that the focus of the paper is to present a practical way in radiograph classification using transfer learning. It would be nice if the authors could provide some future work.

Response: We appreciate that the reviewer has a clear understanding of the ultimate goal of the study. Our future work is currently in progress. We are hoping this paper can lay the foundation as a first step. We have also added a small detail (S3 Fig) which is an image segmentation model that will be used in the future work. In this study, we manually cropped all the images, but this is infeasible for future studies on a larger scale. Therefore, we used manual cropped bounding box/mask to train a model that will allow us to automate image cropping in the future. 

The greater context of this work is to enable larger scale epidemiology studies of bacterial pneumonia in children. Historically, studies that have aspired to do this have used the method for classification of childhood pneumonia radiographs, and employed a team of radiologists with multiple readers per image and an adjudication process to determine a reading. By using such a study to train the algorithm (e.g., PERCH), we intend to release and deploy this tool to facilitate the conduct of similar studies over larger databases of images without the need for a panel of radiologists. Further work is ongoing.

Reviewer #3: Some Recommendations:

Highlight the originality of the paper as model that have been used is already available.

Response: We are not aware of an available deep learning model to classify CXR images in children with pneumonia into WHO-defined pneumonia categories. Also , we would like to clarify that this study is not about developing new architecture, but about finding a practical use case of an available model architecture, as is also pointed out by reviewer 2. We are aware of only one previous study, by Mahomed et al., which was focused on training a model to classify pediatric CXR into WHO-defined pneumonia. We have revised the last paragraph of the introduction to highlight the originality of the paper as compared to the one by Mahomed et al.

Reviewer #3: It is not clear whether cropping of CXRs is done manually or automated. Specify it.(line no 157)

Response: We have clarified cropping process between line 169-173, and added an illustration in S3 Fig. We used manual cropping for this study, and the byproduct of the process is that we have ground-truth label to train a model to automate the cropping process. We have added this detail in the paper now.

Reviewer #3: Discuss the deep-learning model that is used in the study.

Also add the results obtained on various deep-learning models that you have compared.

Response: All the available architectures were widely used in image classification, and the original paper that invented the model has described them in detail. We have now added reference papers for all the models, and readers who are interested in further detail can refer to those papers. 

For the reviewer’s reference, below are the results of all models in a preliminary analysis. A brief summary of the results in the paper is between line 151-155. 

We did not report these results in detail, because typically model architecture selection, like hyperparameter tuning, is a routine process and is not expected to be reported in detail. In previous studies such as https://arxiv.org/pdf/1711.05225.pdf, https://journals.plos.org/plosmedicine/article?id=10.1371/journal.pmed.1002686, https://stanfordmlgroup.github.io/competitions/chexpert/

, the researchers either simply reported the final architecture of selection that yields the lowest loss or highest accuracy in preliminary analysis, or simply mentioned the name of all architectures that were tried. All previous studies found DenseNet 121 to be the optimal architecture for CXR images. 

Reviewer #4: The manuscript is good overall. This study based on deep learning method lays a strong foundation for the potential inclusion of computer-aided pediatric CXR readings in vaccine trials and epidemiology studies. There are some spelling mistakes need to be corrected, such as "trainig" in line 140.

Response: We would like to thank the reviewer for the positive comment. We have proofread the paper again to correct residual spelling errors.

---

## [Decision Letter · Decision Letter 1]

1 Jun 2021

Deep Learning for Classification of Pediatric Chest Radiographs by WHO's Standardized Methodology

PONE-D-21-05523R1

Dear Dr. Chen,

We’re pleased to inform you that your manuscript has been judged scientifically suitable for publication and will be formally accepted for publication once it meets all outstanding technical requirements.

Kind regards,

Khanh N.Q. Le

Academic Editor

PLOS ONE

Additional Editor Comments (optional):

Reviewers' comments:

Reviewer's Responses to Questions

**Comments to the Author**

1. If the authors have adequately addressed your comments raised in a previous round of review and you feel that this manuscript is now acceptable for publication, you may indicate that here to bypass the “Comments to the Author” section, enter your conflict of interest statement in the “Confidential to Editor” section, and submit your "Accept" recommendation.

Reviewer #1: All comments have been addressed

2. Is the manuscript technically sound, and do the data support the conclusions?

Reviewer #1: Yes

3. Has the statistical analysis been performed appropriately and rigorously? 

Reviewer #1: Yes

4. Have the authors made all data underlying the findings in their manuscript fully available?

Reviewer #1: No

5. Is the manuscript presented in an intelligible fashion and written in standard English?

Reviewer #1: Yes

6. Review Comments to the Author

Reviewer #1: The authors have addressed my queries to satisfaction. It is recommended to mention the GitHub/other links to the data and codes that would be populated after possible acceptance of the manuscript.

7. PLOS authors have the option to publish the peer review history of their article (what does this mean?). If published, this will include your full peer review and any attached files.

Reviewer #1: No

---

## [Editor Report · Acceptance letter]

10 Jun 2021

PONE-D-21-05523R1 

Deep Learning for Classification of Pediatric Chest Radiographs by WHO's Standardized Methodology 

Dear Dr. Chen:

I'm pleased to inform you that your manuscript has been deemed suitable for publication in PLOS ONE. Congratulations! Your manuscript is now with our production department. 

Kind regards, 

on behalf of

Dr. Khanh N.Q. Le 

Academic Editor

PLOS ONE